# GENERATIVE PARAGRAPH VECTOR

**Ruqing Zhang, Jiafeng Guo, Yanyan Lan, Jun Xu& Xueqi Cheng**
CAS Key Lab of Network Data Science and Technology
Institute of Computing Technology, Chinese Academy of Sciences
Beijing, China
`zhangruqing@software.ict.ac.cn, {guojiafeng,lanyanyan,junxu,cxq}@ict.ac.cn`

## ABSTRACT

The recently introduced Paragraph Vector is an efficient method for learning high-quality distributed representations for pieces of texts. However, an inherent limitation of Paragraph Vector is lack of ability to infer distributed representations for texts outside of the training set. To tackle this problem, we introduce a Generative Paragraph Vector, which can be viewed as a probabilistic extension of the Distributed Bag of Words version of Paragraph Vector with a complete generative process. With the ability to infer the distributed representations for unseen texts, we can further incorporate text labels into the model and turn it into a supervised version, namely Supervised Generative Paragraph Vector. In this way, we can leverage the labels paired with the texts to guide the representation learning, and employ the learned model for prediction tasks directly. Experiments on five text classification benchmark collections show that both model architectures can yield superior classification performance over the state-of-the-art counterparts.

## 1 INTRODUCTION

A central problem in many text based applications, e.g., sentiment classification (Pang & Lee, 2008), question answering (Stefanie Tellex & Marton., 2003) and machine translation (I. Sutskever & Le, 2014), is how to capture the essential meaning of a piece of text in a fixed-length vector. Perhaps the most popular fixed-length vector representations for texts is the bag-of-words (or bag-of-n-grams) (Harris, 1954). Besides, probabilistic latent semantic indexing (PLSI) (Hofmann, 1999) and latent Dirichlet allocation (LDA) (Blei & Jordan, 2003) are two widely adopted alternatives.

A recent paradigm in this direction is to use a distributed representation for texts (T. Mikolov & Dean, 2013a). In particular, Le and Mikolov (Quoc Le, 2014; Andrew M.Dai, 2014) show that their method, Paragraph Vector (PV), can capture text semantics in dense vectors and outperform many existing representation models. Although PV is an efficient method for learning high-quality distributed text representations, it suffers a similar problem as PLSI that it provides no model on text vectors: it is unclear how to infer the distributed representations for texts outside of the training set with the learned model (*i.e.*, learned text and word vectors). Such a limitation largely restricts the usage of the PV model, especially in those prediction focused scenarios.

Inspired by the completion and improvement of LDA over PLSI, we first introduce the Generative Paragraph Vector (GPV) with a complete generation process for a corpus. Specifically, GPV can be viewed as a probabilistic extension of the Distributed Bag of Words version of Paragraph Vector (PV-DBOW), where the text vector is viewed as a hidden variable sampled from some prior distributions, and the words within the text are then sampled from the softmax distribution given the text and word vectors. With a complete generative process, we are able to infer the distributed representations of new texts based on the learned model. Meanwhile, the prior distribution over text vectors also acts as a regularization factor from the view of optimization, thus can lead to higher-quality text representations.

More importantly, with the ability to infer the distributed representations for unseen texts, we now can directly incorporate labels paired with the texts into the model to guide the representation learning, and turn the model into a supervised version, namely Supervised Generative Paragraph Vector (SGPV). Note that supervision cannot be directly leveraged in the original PV model since it has no

generalization ability on new texts. By learning the SGPV model, we can directly employ SGPV to predict labels for new texts. As we know, when the goal is prediction, fitting a supervised model would be a better choice than learning a general purpose representations of texts in an unsupervised way. We further show that SGPV can be easily extended to accommodate n-grams so that we can take into account word order information, which is important in learning semantics of texts.

We evaluated our proposed models on five text classification benchmark datasets. For the unsupervised GPV, we show that its superiority over the existing counterparts, such as bag-of-words, LDA, PV and FastSent (Felix Hill, 2016). For the SGPV model, we take into comparison both traditional supervised representation models, e.g. MNB (S. Wang, 2012), and a variety of state-of-the-art deep neural models for text classification (Kim, 2014; N. Kalchbrenner, 2014; Socher & Potts, 2013; Irsoy & Cardie, 2014). Again we show that the proposed SGPV can outperform the baseline methods by a substantial margin, demonstrating it is a simple yet effective model.

The rest of the paper is organized as follows. We first review the related work in section 2 and briefly describe PV in section 3. We then introduce the unsupervised generative model GPV and supervised generative model SGPV in section 4 and section 5 respectively. Experimental results are shown in section 6 and conclusions are made in section 7.

## 2 RELATED WORK

Many text based applications require the text input to be represented as a fixed-length feature vector. The most common fixed-length representation is bag-of-words (BoW) (Harris, 1954). For example, in the popular TF-IDF scheme (Salton & McGill, 1983), each document is represented by *tfidf* values of a set of selected feature-words. However, the BoW representation often suffers from data sparsity and high dimension. Meanwhile, due to the independent assumption between words, BoW representation has very little sense about the semantics of the words.

To address this shortcoming, several dimensionality reduction methods have been proposed, such as latent semantic indexing (LSI) (S. Deerwester & Harshman, 1990), Probabilistic latent semantic indexing (PLSI) (Hofmann, 1999) and latent Dirichlet allocation (LDA) (Blei & Jordan, 2003). Both PLSI and LDA have a good statistical foundation and proper generative model of the documents, as compared with LSI which relies on a singular value decomposition over the term-document co-occurrence matrix. In PLSI, each word is generated from a single topic, and different words in a document may be generated from different topics. While PLSI makes great effect on probabilistic modeling of documents, it is not clear how to assign probability to a document outside of the training set with the learned model. To address this issue, LDA is proposed by introducing a complete generative process over the documents, and demonstrated as a state-of-the-art document representation method. To further tackle the prediction task, Supervised LDA (David M.Blei, 2007) is developed by jointly modeling the documents and the labels.

Recently, distributed models have been demonstrated as efficient methods to acquire semantic representations of texts. A representative method is Word2Vec (Tomas Mikolov & Dean, 2013b), which can learn meaningful word representations in an unsupervised way from large scale corpus. To represent sentences or documents, a simple approach is then using a weighted average of all the words. A more sophisticated approach is combing the word vectors in an order given by a parse tree (Richard Socher & Ng, 2012). Later, Paragraph Vector (PV) (Quoc Le, 2014) is introduced to directly learn the distributed representations of sentences and documents. There are two variants in PV, namely the Distributed Memory Model of Paragraph Vector (PV-DM) and the Distributed Bag of Words version of Paragraph Vector (PV-DBOW), based on two different model architectures. Although PV is a simple yet effective distributed model on sentences and documents, it suffers a similar problem as PLSI that it provides no model on text vectors: it is unclear how to infer the distributed representations for texts outside of the training set with the learned model.

Besides these unsupervised representation learning methods, there have been many supervised deep models with directly learn sentence or document representations for the prediction tasks. Recursive Neural Network (RecursiveNN) (Richard Socher & Ng, 2012) has been proven to be efficient in terms of constructing sentence representations. Recurrent Neural Network (RNN) (Ilya Sutskever & Hinton, 2011) can be viewed as an extremely deep neural network with weight sharing across time. Convolution Neural Network (CNN) (Kim, 2014) can fairly determine discriminative phrases in a

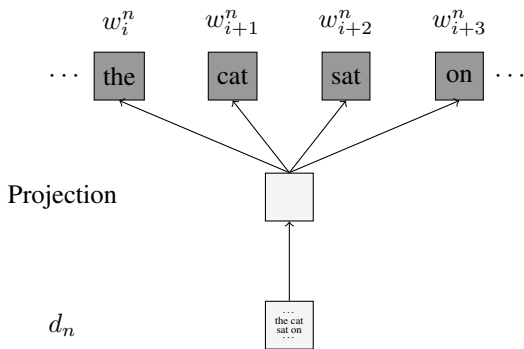

Figure 1: Distributed Bag of Words version of paragraph vectors. The paragraph vector is used to predict the words in a small window ("the", "cat", "sat" and "on").

text with a max-pooling layer. However, these deep models are usually quite complex and thus the training would be time-consuming on large corpus.

## 3 PARAGRAPH VECTOR

Since our model can be viewed as a probabilistic extension of the PV-DBOW model with a complete generative process, we first briefly review the PV-DBOW model for reference.

In PV-DBOW, each text is mapped to a unique paragraph vector and each word is mapped to a unique word vector in a continuous space. The paragraph vector is used to predict target words randomly sampled from the paragraph as shown in Figure 1. More formally, Let $\boldsymbol{D} = \{\boldsymbol{d}_1, \ldots, \boldsymbol{d}_N\}$ denote a corpus of $N$ texts, where each text $\boldsymbol{d}_n = (w_1^n, w_2^n, \ldots, w_{l_n}^n), n \in 1, 2, \ldots, N$ is an $l_n$-length word sequence over the word vocabulary $\boldsymbol{V}$ of size $M$. Each text $\boldsymbol{d} \in \boldsymbol{D}$ and each word $w \in \boldsymbol{V}$ is associated with a vector $\vec{d} \in \mathbb{R}^K$ and $\vec{w} \in \mathbb{R}^K$, respectively, where $K$ is the embedding dimensionality. The predictive objective of the PV-DBOW for each word $w_l^n \in \boldsymbol{d}_n$ is defined by the softmax function

$$p(w_i^n | \boldsymbol{d}_n) = \frac{\exp(\vec{w}_i^n \cdot \vec{d}_n)}{\sum_{w' \in \boldsymbol{V}} \exp(\vec{w}' \cdot \vec{d}_n)} \tag{1}$$

The PV-DBOW model can be efficiently trained using the stochastic gradient descent (Rumelhart & Williams, 1986) with negative sampling (T. Mikolov & Dean, 2013a).

As compared with traditional topic models, e.g. PLSI and LDA, PV-DBOW conveys the following merits. Firstly, PV-DBOW using negative sampling can be interpreted as a matrix factorization over the words-by-texts co-occurrence matrix with shifted-PMI values (Omer Levy & Ramat-Gan, 2015). In this way, more discriminative information (i.e., PMI) can be modeled in PV as compared with the generative topic models which learn over the words-by-texts co-occurrence matrix with raw frequency values. Secondly, PV-DBOW does not have the explicit "topic" layer and allows words automatically clustered according to their co-occurrence patterns during the learning process. In this way, PV-DBOW can potentially learn much finer topics than traditional topic models given the same hidden dimensionality of texts. However, a major problem with PV-DBOW is that it provides no model on text vectors: it is unclear how to infer the distributed representations for unseen texts.

## 4 GENERATIVE PARAGRAPH VECTOR

In this section, we introduce the GPV model in detail. Overall, GPV is a generative probabilistic model for a corpus. We assume that for each text, a latent paragraph vector is first sampled from some prior distributions, and the words within the text are then generated from the normalized exponential (i.e. softmax) distribution given the paragraph vector and word vectors. In our work, multivariate normal distribution is employed as the prior distribution for paragraph vectors. It could

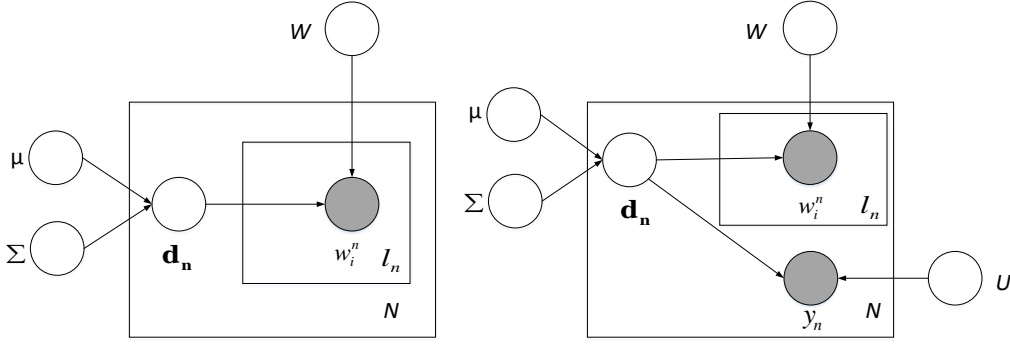

Figure 2: (**Left**) A graphical model representation of Generative Paragraph Vector (GPV). (The boxes are "plates" representing replicates; a shaded node is an observed variable; an unshaded node is a hidden variable.) (**Right**) Graphical model representation of Supervised Generative Paragraph Vector (SGPV).

be replaced by other prior distributions and we will leave this as our future work. The specific generative process is as follows:

For each text $\boldsymbol{d}_n \in \boldsymbol{D}, n = 1, 2, \ldots, N$:

(a) Draw paragraph vector $\vec{d}_n \sim \mathcal{N}(\mu, \Sigma)$

(b) For each word $w_i^n \in \boldsymbol{d}_n, i = 1, 2, \ldots, l_n$ :

Draw word $w_i^n \sim softmax(\vec{d}_n \cdot W)_i$

where $W$ denotes a $k \times M$ word embedding matrix with $W_{*j} = \vec{w}_j$, and $softmax(\vec{d}_n \cdot W)_i$ is the softmax function defined the same as in Equation (1). Figure 2 (**Left**) provides the graphical model of this generative process. Note that GPV differs from PV-DBOW in that the paragraph vector is a hidden variable generated from some prior distribution, which allows us to infer the paragraph vector over future texts given the learned model. Based on the above generative process, the probability of the whole corpus can be written as follows:

$$p(D) = \prod_{n=1}^{N} \int p(\vec{d}_n | \mu, \Sigma) \prod_{w_i^n \in \boldsymbol{d}_n} p(w_i^n | W, \vec{d}_n) d\vec{d}_n$$

To learn the model, direct maximum likelihood estimation is not tractable due to non-closed form of the integral. We approximate this learning problem by using MAP estimates for $\vec{d}_n$, which can be formulated as follows:

$$(\mu^*, \Sigma^*, W^*) = \arg \max_{\mu, \Sigma, W} \prod p(\hat{d}_n | \mu, \Sigma) \prod_{w_i^n \in \boldsymbol{d}_n} p(w_i^n | W, \hat{d}_n)$$

where $\hat{d}_n$ denotes the MAP estimate of $\vec{d}_n$ for $\mathbf{d}_n$, $(\mu^*, \Sigma^*, W^*)$ denotes the optimal solution. Note that for computational simplicity, in this work we fixed $\mu$ as a zero vector and $\Sigma$ as a identity matrix. In this way, all the free parameters to be learned in our model are word embedding matrix $W$. By taking the logarithm and applying the negative sampling idea to approximate the softmax function, we obtain the final learning problem

$$\mathcal{L} = \sum_{n=1}^{N} \left( -\frac{1}{2} ||\hat{d}_n||^2 + \sum_{w_i^n \in \boldsymbol{d}_n} \left( \log \sigma(\vec{w}_i^n \cdot \hat{d}_n) + k \cdot \mathbb{E}_{w' \sim P_{nw}} \log \sigma(-\vec{w'} \cdot \hat{d}_n) \right) \right)$$

where $\sigma(x) = 1/(1 + \exp(-x))$, $k$ is the number of "negative" samples, $w'$ denotes the sampled word and $P_{nw}$ denotes the distribution of negative word samples. As we can see from the final objective function, the prior distribution over paragraph vectors actually act as a regularization term. From the view of optimization, such regularization term could constrain the learning space and usually produces better paragraph vectors.

For optimization, we use coordinate ascent, which first optimizes the word vectors $W$ while leaving the MAP estimates ($\hat{d}$) fixed. Then we find the new MAP estimate for each document while leaving the word vectors fixed, and continue this process until convergence. To accelerate the learning, we adopt a similar stochastic learning framework as in PV which iteratively updates $W$ and estimates $\vec{d}$ by randomly sampling text and word pairs.

At prediction time, given a new text, we perform an inference step to compute the paragraph vector for the input text. In this step, we freeze the vector representations of each word, and apply the same MAP estimation process of $\vec{d}$ as in the learning phase. With the inferred paragraph vector of the test text, we can feed it to other prediction models for different applications.

## 5    SUPERVISED GENERATIVE PARAGRAPH VECTOR

With the ability to infer the distributed representations for unseen texts, we now can incorporate the labels paired with the texts into the model to guide the representation learning, and turn the model into a more powerful supervised version directly towards prediction tasks. Specifically, we introduce an additional label generation process into GPV to accommodate text labels, and obtain the Supervised Generative Paragraph Vector (SGPV) model. Formally, in SGPV, the $n$-th text $\boldsymbol{d}_n$ and the corresponding class label $y_n \in \{1, 2, \ldots, C\}$ arise from the following generative process:

For each text $\boldsymbol{d}_n \in \boldsymbol{D}, n = 1, 2, \ldots, N$:

(a) Draw paragraph vector $\vec{d}_n \sim \mathcal{N}(\mu, \Sigma)$

(b) For each word $w_i^n \in \boldsymbol{d}_n, i = 1, 2, \ldots, l_n$ :

   Draw word $w_i^n \sim softmax(\vec{d}_n \cdot W)_i$

(c) Draw label $y_n | \vec{d}_n, U, b \sim softmax(U \cdot \vec{d}_n + b)$

where $U$ is a $C \times K$ matrix for a dataset with $C$ output labels, and $b$ is a bias term.

The graphical model of the above generative process is depicted in Figure 2 (**Right**). SGPV defines the probability of the whole corpus as follows

$$p(D) = \prod_{n=1}^{N} \int p(\vec{d}_n | \mu, \Sigma) \Big( \prod_{w_i^n \in \boldsymbol{d}_n} p(w_i^n | W, \vec{d}_n) \Big) p(y_n | \vec{d}_n, U, b) d\vec{d}_n$$

We adopt a similar learning process as GPV to estimate the model parameters. Since the SGPV includes the complete generative process of both paragraphs and labels, we can directly leverage it to predict the labels of new texts. Specifically, at prediction time, given all the learned model parameters, we conduct an inference step to infer the paragraph vector as well as the label using MAP estimate over the test text.

The above SGPV may have limited modeling ability on text representation since it mainly relies on uni-grams. As we know, word order information is often critical in capturing the meaning of texts. For example, "machine learning" and "learning machine" are totally different in meaning with the same words. There has been a variety of deep models using complex architectures such as convolution layers or recurrent structures to help capture such order information at the expense of large computational cost.

Here we propose to extend SGPV by introducing an additional generative process for n-grams, so that we can incorporate the word order information into the model and meanwhile keep its simplicity in learning. We name this extension as SGPV-ngram. Here we take the generative process of SGPV-bigram as an example.

For each text $\boldsymbol{d}_n \in \boldsymbol{D}, n = 1, 2, \ldots, N$:

(a) Draw paragraph vector $\vec{d}_n \sim \mathcal{N}(\mu, \Sigma)$

(b) For each word $w_i^n \in \boldsymbol{d}_n, i = 1, 2, \ldots, l_n$ :

   Draw word $w_i^n \sim softmax(\vec{d}_n \cdot W)_i$

(c) For each bigram $g_i^n \in \boldsymbol{d}_n, i = 1, 2, \ldots, s_n$ :

Draw bigram $g_i^n \sim softmax(\vec{d}_n \cdot G)_i$

(d) Draw label $y_n | \vec{d}_n, U, b \sim softmax(U \cdot \vec{d}_n + b)$

where $G$ denotes a $K \times S$ bigram embedding matrix with $G_{*j} = \vec{g}_j$, and $S$ denotes the size of bigram vocabulary. The joint probability over the whole corpus is then defined as

$$p(D) = \prod_{n=1}^{N} \int p(\vec{d}_n | \mu, \Sigma) \big( \prod_{w_i^n \in \boldsymbol{d}_n} p(w_i^n | W, \vec{d}_n) \big) \big( \prod_{g_i^n \in \boldsymbol{d}_n} p(g_i^n | G, \vec{d}_n) \big) p(y_n | \vec{d}_n, U, b) d\vec{d}_n$$

## 6 EXPERIMENTS

In this section, we introduce the experimental settings and empirical results on a set of text classification tasks.

### 6.1 DATASET AND EXPERIMENTAL SETUP

We made use of five publicly available benchmark datasets in comparison.

**TREC**: The TREC Question Classification dataset (Li & Roth, 2002)[1] which consists of $5,452$ train questions and $500$ test questions. The goal is to classify a question into 6 different types depending on the answer they seek for.

**Subj**: Subjectivity dataset (Pang & Lee, 2004) which contains $5,000$ subjective instances and $5,000$ objective instances. The task is to classify a sentence as being subjective or objective.

**MR**: Movie reviews (Pang & Lee, 2005) [2] with one sentence per review. There are $5,331$ positive sentences and $5,331$ negative sentences. The objective is to classify each review into positive or negative category.

**SST-1**: Stanford Sentiment Treebank (Socher & Potts, 2013) [3]. SST-1 is provided with train/dev/test splits of size $8,544/1,101/2,210$. It is a fine-grained classification over five classes: very negative, negative, neutral, positive, and very positive.

**SST-2**: SST-2 is the same as SST-1 but with neutral reviews removed. We use the standard train/dev/test splits of size $6,920/872/1,821$ for the binary classification task.

Preprocessing steps were applied to all datasets: words were lowercased, non-English characters and stop words occurrence in the training set are removed. For fair comparison with other published results, we use the default train/test split for TREC, SST-1 and SST-2 datasets. Since explicit split of train/test is not provided by subj and MR datasets, we use 10-fold cross-validation instead.

In our model, text and word vectors are randomly initialized with values uniformly distributed in the range of [-0.5, +0.5]. Following the practice in (Tomas Mikolov & Dean, 2013b) , we set the noise distributions for context and words as $p_{\mathrm{nw}}(w) \propto \#(w)^{0.75}$. We adopt the same linear learning rate strategy where the initial learning rate of our models is 0.025. For unsupervised methods, we use support vector machines (SVM) [4] as the classifier.

### 6.2 BASELINES

We adopted both unsupervised and supervised methods on text representation as baselines.

#### 6.2.1 UNSUPERVISED BASELINES

**Bag-of-word-TFIDF and Bag-of-bigram-TFIDF**. In the bag-of-word-TFIDF scheme (Salton & McGill, 1983) , each text is represented as the *tf-idf* value of chosen feature-words. The bag-of-

---

[1] http://cogcomp.cs.illinois.edu/Data/QA/QC/
[2] https://www.cs.cornell.edu/people/pabo/movie-review-data/
[3] http://nlp.stanford.edu/sentiment/
[4] http://www.csie.ntu.edu.tw/~cjlin/libsvm/

bigram-TFIDF model is constructed by selecting the most frequent unigrams and bigrams from the training subset. We use the vanilla TFIDF in the gensim library[5].

**LSI** (S. Deerwester & Harshman, 1990) and **LDA** (Blei & Jordan, 2003). LSI maps both texts and words to lower-dimensional representations in a so-called latent semantic space using SVD decomposition. In LDA, each word within a text is modeled as a finite mixture over an underlying set of topics. We use the vanilla LSI and LDA in the gensim library with topic number set as 100.

**cBow** (Tomas Mikolov & Dean, 2013b). Continuous Bag-Of-Words model. We use average pooling as the global pooling mechanism to compose a sentence vector from a set of word vectors.

**PV** (Quoc Le, 2014). Paragraph Vector is an unsupervised model to learn distributed representations of words and paragraphs.

**FastSent** (Felix Hill, 2016). In FastSent, given a simple representation of some sentence in context, the model attempts to predict adjacent sentences.

Note that unlike LDA and GPV, LSI, cBow, and FastSent cannot infer the representations of unseen texts. Therefore, these four models need to fold-in all the test data to learn representations together with training data, which makes it not efficient in practice.

### 6.2.2 SUPERVISED BASELINES

**NBSVM** and **MNB** (S. Wang, 2012). Naive Bayes SVM and Multinomial Naive Bayes with unigrams and bi-grams.

**DAN** (Mohit Iyyer & III, 2015). Deep averaging network uses average word vectors as the input and applies multiple neural layers to learn text representation under supervision.

**CNN-multichannel** (Kim, 2014). CNN-multichannel employs convolutional neural network for sentence modeling.

**DCNN** (N. Kalchbrenner, 2014). DCNN uses a convolutional architecture that replaces wide convolutional layers with dynamic pooling layers.

**MV-RNN** (Richard Socher & Ng, 2012). Matrix-Vector RNN represents every word and longer phrase in a parse tree as both a vector and a matrix.

**DRNN** (Irsoy & Cardie, 2014). Deep Recursive Neural Networks is constructed by stacking multiple recursive layers.

**Dependency Tree-LSTM** (Kai Sheng Tai & Manning, 2015). The Dependency Tree-LSTM based on LSTM structure uses dependency parses of each sentence.

### 6.3 PERFORMANCE OF GENERATIVE PARAGRAPH VECTOR

We first evaluate the GPV model by comparing with the unsupervised baselines on the TREC, Subj and MR datasets. As shown in table 1, GPV works better than PV over the three tasks. It demonstrates the benefits of introducing a prior distribution (i.e., regularization) over the paragraph vectors. Moreover, GPV can also outperform almost all the baselines on three tasks except Bow-TFIDF and Bigram-TFIDF on the TREC collection. The results show that for unsupervised text representation, bag-of-words representation is quite simple yet powerful which can beat many embedding models. Meanwhile, by using a complete generative process to infer the paragraph vectors, our model can achieve the state-of-the-art performance among the embedding based models.

### 6.4 PERFORMANCE OF SUPERVISED GENERATIVE PARAGRAPH VECTOR

We compare SGPV model to supervised baselines on all the five classification tasks. Empirical results are shown in Table 2. We can see that SGPV achieves comparable performance against other deep learning models. Note that SGPV is much simpler than these deep models with significantly less parameters and no complex structures. Moreover, deep models with convolutional layers or recurrent structures can potentially capture compositional semantics (e.g., phrases), while SGPV only

---

[5] http://radimrehurek.com/gensim/

Table 1: Performance Comparison of Unsupervised Representation Models.

| Model | TREC | Subj | MR |
|---|---|---|---|
| BoW-TFIDF | 97.2 | 89.8 | 76.7 |
| Bigram-TFIDF | 97.6 | 90.9 | 76.1 |
| LSI | 88 | 85.4 | 64.2 |
| LDA | 81.3 | 71 | 61.6 |
| cBow (Han Zhao & Poupart, 2015) | 87.3 | 91.3 | 77.2 |
| PV (Han Zhao & Poupart, 2015) | 91.8 | 90.5 | 74.8 |
| FastSent (Felix Hill, 2016) | 76.8 | 88.7 | 70.8 |
| GPV | 93 | 91.7 | 77.9 |

relies on uni-gram. In this sense, SGPV is quite effective in learning text representation. Meanwhile, if we take Table 1 into consideration, it is not surprising to see that SGPV can consistently outperform GPV on all the three classification tasks. This also demonstrates that it is more effective to directly fit supervised representation models than to learn a general purpose representation in prediction scenarios.

By introducing bi-grams, SGPV-bigram can outperform all the other deep models on four tasks. In particular, the improvements of SGPV-bigram over other baselines are significant on SST-1 and SST-2. These results again demonstrated the effectiveness of our proposed SGPV model on text representations. It also shows the importance of word order information in modeling text semantics.

Table 2: Performance Comparison of Supervised Representation Models.

| Model | SST-1 | SST-2 | TREC | Subj | MR |
|---|---|---|---|---|---|
| NBSVM (S. Wang, 2012) | - | - | - | 93.2 | 79.4 |
| MNB (S. Wang, 2012) | - | - | - | 93.6 | 79 |
| DAN (Mohit Iyyer & III, 2015) | 47.7 | 86.3 | - | - | - |
| CNN-multichannel (Kim, 2014) | 47.4 | 88.1 | 92.2 | 93.2 | 81.1 |
| DCNN (N. Kalchbrenner, 2014) | 48.5 | 86.8 | 93 | - | - |
| MV-RNN (Richard Socher & Ng, 2012) | 44.4 | 82.9 | - | - | 79 |
| DRNN (Irsoy & Cardie, 2014) | 49.8 | 86.6 | - | - | - |
| Dependency Tree-LSTM (Kai Sheng Tai & Manning, 2015) | 48.4 | 85.7 | - | - | - |
| SGPV | 44.6 | 86.3 | 93.2 | 92.4 | 79.2 |
| SGPV-bigram | 55.9 | 91.8 | 95.8 | 93.6 | 79.8 |

## 7    CONCLUSIONS

In this paper, we introduce GPV and SGPV for learning distributed representations for pieces of texts. With a complete generative process, our models are able to infer vector representations as well as labels over unseen texts. Our models keep as simple as PV models, and thus can be efficiently learned over large scale text corpus. Even with such simple structures, both GPV and SGPV can produce state-of-the-art results as compared with existing baselines, especially those complex deep models. For future work, we may consider other probabilistic distributions for both paragraph vectors and word vectors.

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
