# Peer review of "Generative Paragraph Vector"

_ICLR 2017 — rejected_

[Official Review · AnonReviewer1 · rating 2 · confidence 5 · 17 Dec 2016]
**Very limited in novelty**

This work reframes paragraph vectors from a generative point of view and in so doing, motivates the existing method of inferring paragraph vectors as well as applying a L2 regularizer on the paragraph embeddings. The work also motivates joint learning of a classifier on the paragraph vectors to perform text classification.

The paper has numerous citation issues both in formatting within the text and the formatting of the bibliography, e.g. on some occasions including first names, on others not. I suggest the authors use a software package like BibTex to have a more consistent bibliography. There seems to be little novelty in this work. 

The authors claim that there is no proposed method for inferring unseen documents for paragraph vectors. This is untrue. In the original paragraph vector paper, the authors show that to get a new vector, the rest of the model parameters are held fixed and gradient descent is performed on the new paragraph vector. This means the original dataset is not needed when inferring a paragraph vector for new text. This work seems to be essentially doing the same thing when finding the MAP estimate for a new vector. Thus the only contribution from the generative paragraph vector framing is the regularization on the embedding matrix.

The supervised generative paragraph vector amounts to jointly training a linear classifier on the paragraph vectors, while inference for the paragraph vector is unchanged. For the n-gram based approach, the authors should cite Li et al., 2015.

In the experiments, table 1 and 2 are badly formatted with .0 being truncated. The authors also do not state the size of the paragraph vector. Finally the SGPV results are actually worse than that reported in the original paragraph vector paper where SST-1 got 48.7 and SST-2 got 86.3.

Bofang Li, Tao Liu, Xiaoyong Du, Deyuan Zhang, Zhe Zhao, Learning Document Embeddings by Predicting N-grams for Sentiment Classification of Long Movie Reviews, 2015.

[Official Review · AnonReviewer2 · rating 3 · confidence 4 · 19 Dec 2016]
**Not convincing**

It feels that this paper is structured around a shortcoming of the original paragraph vectors paper, namely an alleged inability to infer representation for text outside of the training data. I am reasonably sure that this is not the case. Unfortunately on that basis, the premise for the work presented here no longer holds, which renders most of the subsequent discussion void.

While I recommend this paper be rejected, I encourage the authors to revisit the novel aspects of the idea presented here and see if that can be turned into a different type of paper going forward.

[Official Review · AnonReviewer3 · rating 4 · confidence 4 · 24 Dec 2016]
**below borderline**

While this paper has some decent accuracy numbers, it is hard to argue for acceptance given the following:

1) motivation based on the incorrect assumption that the Paragraph Vector wouldn't work on unseen data
2) Numerous basic formatting and Bibtex citation issues.

Lack of novelty of yet another standard directed LDA-like bag of words/bigram model.

[Final Decision · Program Chairs · 06 Feb 2017]
**ICLR committee final decision**

The contribution of this paper generally boils down to adding a prior to the latent representations of the paragraph in the Paragraph Vector model. An especially problematic point about this paper is the claim that the original paper considered only the transductive setting (i.e. it could not induce representations of new documents). It is not accurate, they also used gradient descent at test time. Though I agree that regularizing the original model is a reasonable thing to do, I share the reviewers' feeling that the contribution is minimal. There are also some serious issues with presentation (as noted by the reviewers), I am surprised that the authors have not addressed them during the review period.